# A digitalized program to improve antenatal health care in a rural setting in North-Western Burundi: Early evidence-based lessons

**Nadine Misago**[1]*, **Desire Habonimana**[2], **Roger Ciza**[1], **Jean Paul Ndayizeye**[3], **Joyce Kevin Abalo Kimaro**[4]

**1** Health Healing Network Burundi, Santé Maternelle et Néonatale, Bujumbura, Burundi, **2** Centre de Recherche Universitaire en Santé (CURSA), Department of Community Medicine, Faculty of Medicine, University of Burundi, Bujumbura, 5190, Burundi, **3** Department of Community Medicine, University of Global Health Equity, Butaro, Rwanda, **4** East African Market Driven and People Centred Integration, Incubator for Integration and Development in East Africa, Deutsche Gesellschaft für Internationale Zusammenarbeit (GIZ) GmbH, East Africa Community, Arusha, Tanzania

* missnad1988@gmail.com

**Data Availability Statement:** Data and STATA commands (dofile) used in this manuscript are publicly available on Figshare. https://doi.org/10.6084/m9.figshare.21828066.v3.

## Abstract

In Burundi, the north-western region continues to grapple with the lowest level of antenatal care (ANC) attendance rate which is constantly about half the national average of 49% ANC4 coverage. Despite a dearth of empirical evidence to understand the determinants of this suboptimal attendance of ANC, widespread evidence informs that women forget scheduled ANC appointments. We designed and tested a digital intervention that uses a reminder model aimed at increasing the number of women who attend at least 4 ANC visits in this region. We enrolled a cohort of 132 pregnant women who were followed until childbirth using a single arm pre- and post-test design. The digital model builds on the collaboration between midwives or nurses, community health workers (CHWs), and pregnant women who are centrally connected through regular automated communications generated by the cPanel of the digital intervention. In addition to ANC attendances, we nested a cross-sectional survey to understand mothers' perceptions and acceptability of the digital intervention using the acceptability framework by Sekhon et al. (2017). Descriptive analyses were performed to observe the trend in ANC attendance and logistic regressions fitted to seize determinants affecting mothers' acceptability of the intervention. Of 132 enrolled pregnant women, 1 (0.76%) dropped out. From a baseline of 23%, nearly 73.7% of mothers attended their subsequent ANC visits after the start of the intervention. From the third month of intervention, about 80% of mothers constantly attended ANC appointments; which corresponds to greater than 200% increase from the baseline. Findings showed that 96.2% of mothers expressed satisfaction, 77.1% positively reacted to automated reminders (attitudes), 70.2% expressed willingness to participate, and 86.3% had the ability to actively participate to the intervention. Conversely, half of mothers confirmed that participation to this programme somewhat affected their time management. A key learning is that digital interventions have a lot of promise to improve pregnancy monitoring in rural settings. However, the overall user acceptability was low especially among mothers lacking personal mobile phone.

**Funding:** The study was funded by The Deutsche Gesellschaft für Internationale Zusammenarbeit (GIZ) through The Incubator for Integration and Development in East Africa (IIDEA) program of The East Africa Community. The funders had no role in study design, data collection and analysis, decision to publish, or preparation of the manuscript.

## Author summary

Digital interventions have showed a lot of promise in improving health outcomes in many settings. In the north-western region of Burundi, antenatal care (ANC) attendance is significantly low compared to the countrywide levels. Our study tested a small-scale digital intervention on rural pregnant women in this region with an aim to increase ANC attendance. The intervention was built on a reminder model using mobile phones and involved collaboration of pregnant women, community health workers (CHWs), and health care providers namely midwives and nurses. Findings support the evidence that mobile-based interventions can improve pregnancy monitoring in rural settings. However, mothers' acceptability of the digital health interventions is low mainly among those not owning a mobile phone. Considering that mobile phone ownership, including android-based mobile devices, is rapidly expanding; this constitutes an asset for scaling up innovative digital health interventions to improve health outcomes. Engagement of local players by using a bottom-up approach is paramount and should guide future community-targeted digital health interventions.

## Introduction

Maternal and newborn's health (MNH) is a priority goal as enshrined in the Sustainable Development Goals (SDGs) agenda 2016–2030 [1] and specifically emphasized through the Global Strategy for Women's, Children's and Adolescents' Health agenda 2016–2030 [2]. The global agenda set a bold ambition to drop maternal mortality ratio below 70 deaths per 100,000 live-births and further reduce neonatal mortality below 12 deaths per 1,000 livebirths by 2030 [3,4]. Despite the global decline in maternal and neonatal deaths over the past millennium agenda, some countries like Burundi still grapples with high rates of these deaths. In 2017 for instance, Burundi continues to register higher mortality ratios (334 maternal deaths per 100,000 live-births and 23 neonatal deaths per 1,000 livebirths in 2017) [5]. In line with the global targets, Burundi set a bid to reduce maternal mortality by 58% (attain at most 140 maternal deaths per 100,000 livebirths) and further halve neonatal mortality (achieve 12 deaths per 1,000 livebirths) by 2030 [6]. To attain these mortality goals by the deadline, the country needs to invest in quality maternal care services spanning the pregnancy, childbirth, and postpartum periods. Our study focused on the pregnancy period and sought to increase the number of pregnant women who attend scheduled ANC appointments.

Evidence has shown that timely and quality ANC improves women's pregnancy experience and averts many maternal and neonatal health complications including deaths [7–9]. To support the preceding, mounting evidence has established a correlation between ANC visits and maternal and neonatal health outcomes. For instance, women who use ANC services are more likely to deliver in a health facility which constitutes cornerstone of better health outcomes for the mother and the newborn [10,11]. Most evidently, in a recent nationally representative study to understand the socio-economic determinants of maternal health services utilization in Burundi, Habonimana et al. (2021) found that women who completed four or more ANC visits are 14 times more likely to have an assisted childbirth [11]. Unlike those having assisted childbirth in a health setting, women who deliver outside a health setting are likely to experience risky circumstances and are further exposed to fatal outcomes especially if an emergency complication such as severe bleeding or the need for an urgent c-section arises [12–14].

Therefore, with an aim to improve maternal and newborn health, the World Health Organization (WHO) recommends that pregnant women have eight contacts with a skilled health care provider during the course of the pregnancy period, delivers in a health facility setting, and further be assisted by a skilled birth attendant during childbirth [15,16]. However, despite the Government of Burundi having fully subsidized maternal and child health services especially ANC which is the cornerstone of maternal and newborn health outcome, the nationally representative Demographic and Health Survey (DHS) conducted in 2017 in Burundi revealed that only 49% pregnant women attend four ANC visits [17]. Antenatal care attendance rate is even lower in rural settings, averagely falling below 37% [18,19]. Markedly, the rural setting of the north-western Burundi suffers extremely low rate of ANC attendance with less than 24% of pregnant women attending at least four ANC visits. The low level of education of mothers in low- and middle-income countries (LMICs), the lack of full awareness of the importance of ANC services, and the seldom presence of effective systems to remind mothers of ANC appointments pose a serious problem to the success of maternal and newborn health initiatives in these countries [20]. It is against this rationale that many public health players introduced digital interventions aimed to improve health services consumption by increasing user attendance rate which was achieved by implementing reminder systems [21–23]. Digital interventions have proved successful in increasing service coverage in many contexts. For instance, in a systemic review that sought to gather evidence on the effectiveness of mHealth interventions in increasing service coverage for ANC, postnatal care (PNC), and child immunization in LMICs, all included studies demonstrated that these digital tools are associated with better outcomes [24].

We aimed to test a mobile-based intervention that seeks to improve ANC attendance in the rural north-western setting in Burundi. The intervention's main hypothesis is that the digitalization of ANC program to generate and send automated reminders for scheduled ANC appointments is associated with an increase in ANC attendance. Furthermore, the intervention supported live communication between healthcare providers, community health workers (CHWs), and pregnant women to foster immediate and timely feedback on any pregnancy-related problem. The program was supported by Deutsche Gesellschaft für Internationale Zusammenarbeit (GIZ) and the Incubator for Integration and Development in East Africa (IIDEA) and piloted by a local public health non-governmental organization called Health Healing Network Burundi (HHNB). Murwi health center, a rural clinic located in north-western Burundi served as the pilot setting. In this setting, ANC attendance rate was reported to be below half the national average as of 2017 [17]. Furthermore, prior to intervention, we conducted a six-month baseline follow-up of routine monthly ANC data and found that only 23.2% of women attended at least 4 ANC appointments. Specifically, pre-intervention data showed that 36.2% women attended the first ANC visit and only 4.1% attended the second ANC visit. Most importantly, despite the nearing of delivery, still less than 30% and 24% women attended the third and fourth ANC visits, respectively (**Fig 1**). According to baseline findings, majority of women who missed scheduled ANC appointments confirmed that they forgot these appointments and did not receive any follow-up communication from the health care provider or the CHW to remind them about upcoming appointments or reschedule missed visits. Therefore, baseline findings corroborated with existing evidence and further informed the intervention design. The design of the intervention used a collaborative and bottom-up approach by involving pregnant women, CHWs, midwives and nurses, and by seeking support and buy-in from the funders and the National Programme on Reproductive, Maternal and Newborn Health.

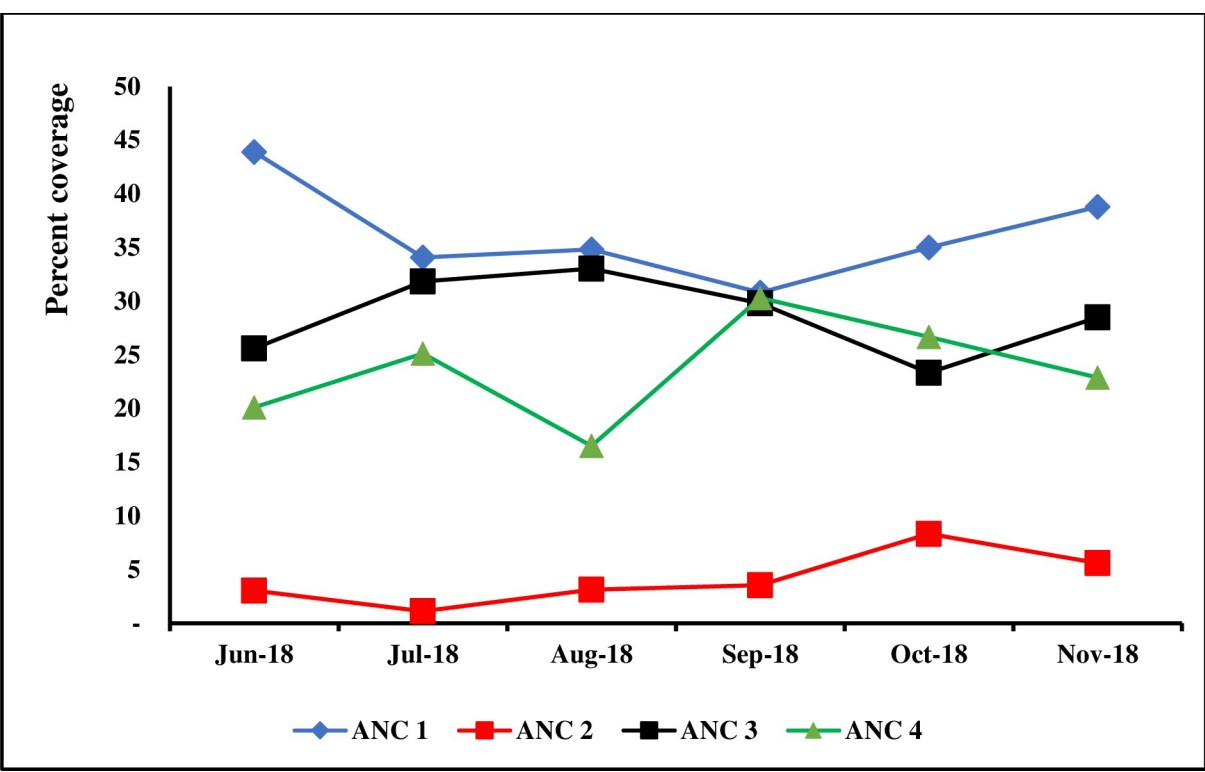

**Fig 1. Baseline ANC attendance rate in Murwi Health Centre.** Fig 1 is a graphical depiction of ANC attendance rates before intervention. Data on the first ANC visit (blue line), second ANC visit (red line), third ANC visit (black line) and fourth ANC visit (green line) were collected for a period of six months preceding intervention. Averages were computed and depicted on a run chart to observe the baseline trend.

## Methods

### Study description

We designed an uncontrolled single-arm pre- and -post study with a six-month baseline follow-up, 12-month intervention, and an endline cross-sectional evaluation. During the baseline, we collected data on ANC attendance in the study setting and used a run-chart to trace trends in ANC1, ANC2, ANC3, and ANC4 attendances (**Fig 1**). Thereafter, we recruited 132 pregnant women using an open cohort design over a period of three months of recruitment. All pregnant women in the health facility catchment area were included provided that they consent to participate. Participating women were followed up until delivery. We collected data on ANC attendances using the cPanel (**Fig 2**) and nested an endline cross-sectional survey to understand user acceptability of the intervention and factors affecting this acceptability.

### Intervention

The intervention, called "Mobile Platform for Maternal Health", consisted of a digital-based innovation that is compatible with both the simple and android mobile-based applications. The platform allows nurses and midwives and CHWs to track pregnant women's ANC appointments, follow up ANC visits and services utilisation and permits interactive information sharing amongst enrolled pregnant women, CHWs, and nurses and midwives. The platform was translated into local language (Kirundi) to enable comprehension. As shown in **Fig 2**, the digital intervention is based on five pathways connecting the cPanel, health care

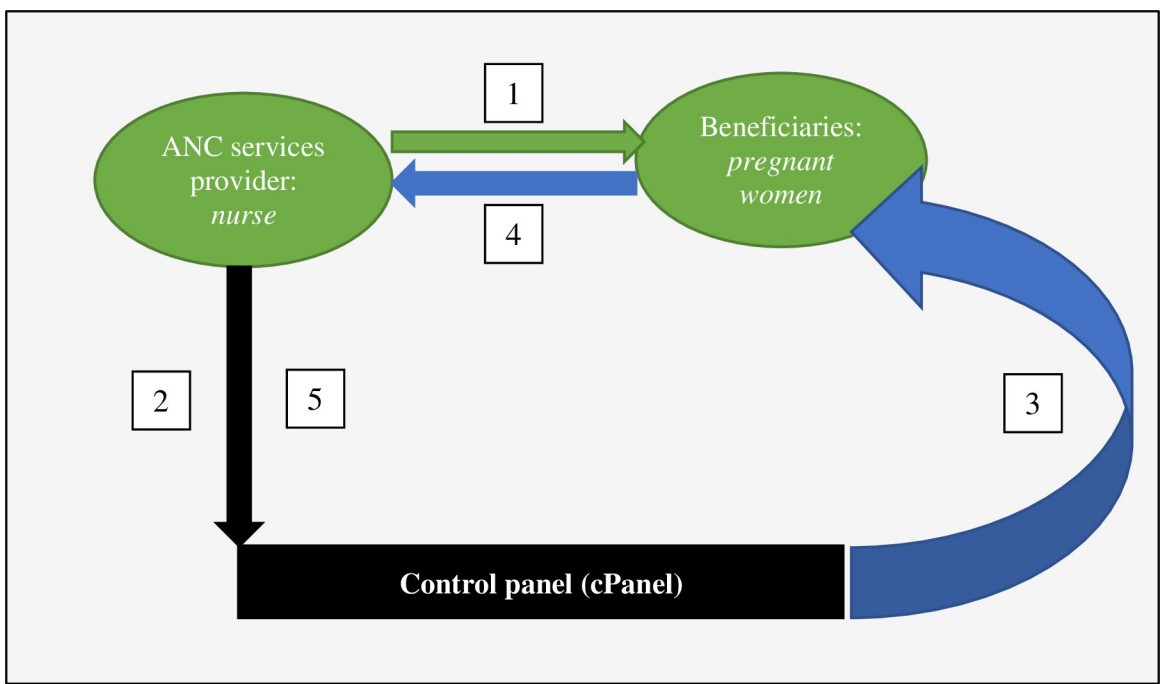

**Fig 2. Digitalized ANC intervention.** In the first instance, a nurse or midwife schedules a pregnant woman for ANC visit. Immediately, both the pregnant woman (1) and the cPanel (2) receive schedule notification as "pop-up message". In the second time, the cPanel sends automated ANC reminders to the pregnant woman (3). The first reminder was generated seven days before, the second three days before, and the third reminder the day before a scheduled appointment. When a woman attends the scheduled ANC appointment (4), the nurse confirms the show-up which is captured by the cPanel (5). The digital platform had an option for live chat and direct free call between nurses and women. Any data going through was stored by the cPanel.

providers (nurses and midwives), and community-based participants (pregnant women and CHWs). We did not provide motivation or incentives for participation.

## Study outcomes

The ultimate study outcome was ANC attendances. The aim of the intervention was to increase ANC attendances up to at least 4 ANC visits as recommended for a better pregnancy experience and an enhanced maternity experience. We used the cPanel to track ANC1, ANC2, ANC3, and ANC4 visits. We also measured acceptability of the intervention by pregnant women which was done using an endline cross-sectional survey on participating mothers. In the light of existing evidence such as the Zambia child immunization program, consumer acceptability determines the uptake of a public health intervention [25].

## Data analysis

From the time of enrolment, we recorded each ANC visit of each pregnant woman. We calculated the proportions of women who attended the first ANC visit, the second ANC visit, and the third and fourth ANC visits; respectively. Proportions were plotted on a run chart to observe the trend compared to the baseline ANC attendance levels. Next, the analysis of cross-sectional survey data aimed at understanding the acceptability by using the healthcare interventions acceptability framework by Sekhon et al. (2017). We adapted the acceptability framework by Sekhon et al. (2017) and created five constructs namely satisfaction, attitudes, willingness, ability, and side effects as shown in **Fig 3** and **Table 1** [26]. In the first instance,

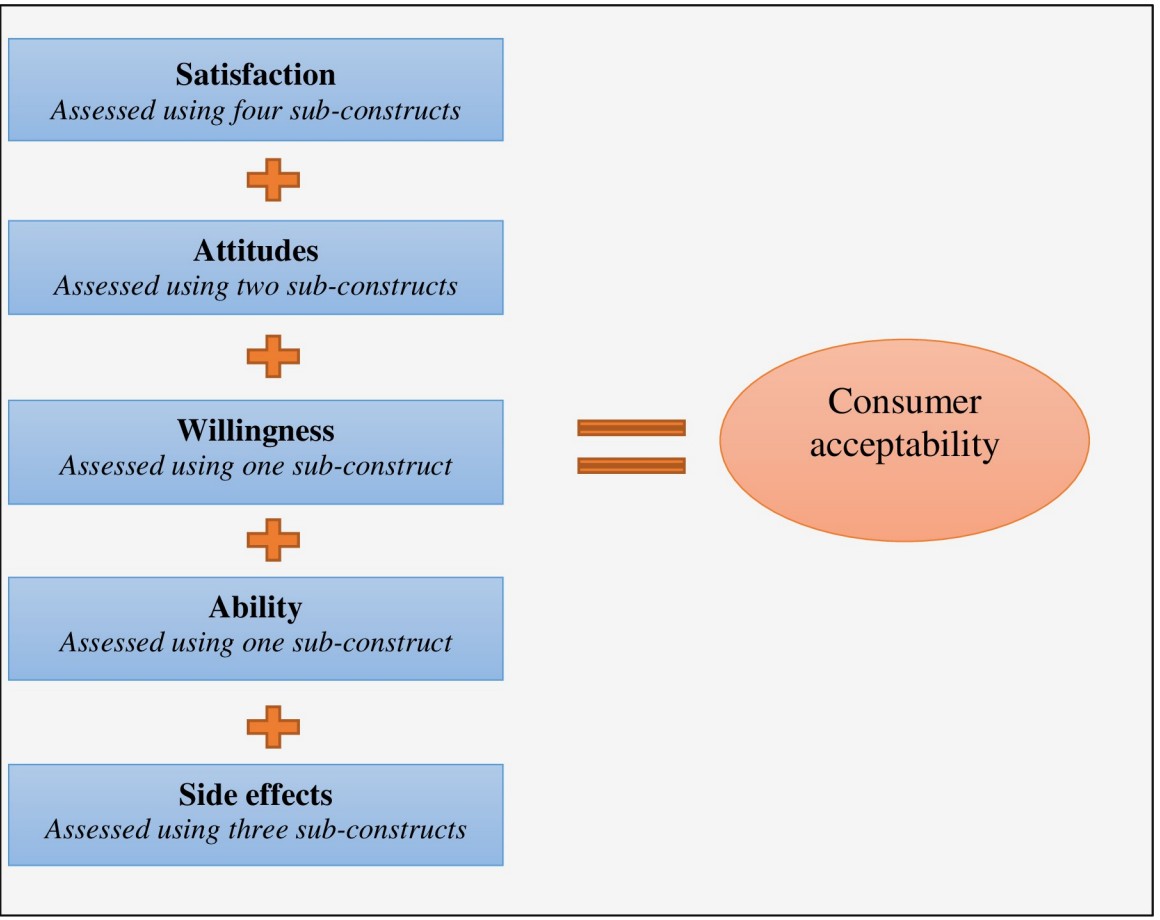

**Fig 3. Consumer acceptability framework.**

constructs were measured independently and then summed up to yield overall acceptability. In the second instance, acceptability was coded as a binary outcome taking value **1** if mother *i* accepted the intervention and value **0**, otherwise. The coding of the outcome variable required a bit of work because it is a composite outcome constructed based on various inputs which were either binary or scale measurements. Binary variables were coded **1** if the response is 'Yes' and **0** otherwise. Similarly, we set a cut-off for scale measurements, taking value **1** for positive responses and **0**, otherwise. The outcome binary variable was obtained by combining construct measures. A logistic regression was fitted to determine factors affecting acceptability. The study received ethics clearance from the Ethics Committee of the school of Medicine of the University of Burundi (Ref. FM/CE03/2020).

## Results

### Antenatal care attendance

As summarized on **Fig 4** below, the intervention created an impact. In fact, 73.7% women attended a follow-up ANC visit a month after the start of the intervention. From the third month of intervention, nearly 80% of women constantly attended subsequent ANC appointments; which represents more than 200% increase from 23.2% pre-intervention ANC attendance rate. The trend lines look stable before and during intervention despite a much bigger

**Table 1. Acceptability constructs of the ANC digital intervention.**

| Acceptability construct and questions |
| --- |
| **Satisfaction (n = 4)** |
| *The extent to which pregnant women feel comfortable with the digital intervention* |
| 1. How comfortable did you feel when you received the first ANC appointment reminder message? |
| 2. How comfortable did you feel when you received subsequent reminder messages? |
| 3. How comfortable did you feel to communicate your appointment with your partner, CHW, or nurse? |
| 4. How comfortable did you feel to attend the ANC appointment based on the reminder messages? |
| **Attitudes (n = 2)** |
| *The expression of pregnant women's behaviour with respect to their participation to the program* |
| 1. How much time did it take you to read the ANC appointment reminder message after it popped up? |
| 2. Did you communicate or note your appointment date based on the reminder message? |
| **Willingness (n = 1)** |
| *The willingness of pregnant women to continue the intervention or recommend its scale up* |
| 1. Would you wish to receive similar reminder if you were to become pregnant in the future? |
| **Ability or self-efficacy (n = 1)** |
| *The level of confidence about engaging in the program without needing external support* |
| 1. Did you seek help to engage in any of the intervention aspects (e.g., borrowing a mobile phone, reading reminder messages, communicating the CHW or nurse, etc.)? |
| **Side effects (n = 3)** |
| *How pregnant women perceives the intervention and its effects on the overall wellbeing* |
| 1. On a scale of 1 to 5, rate the burdensomeness of this intervention |
| 2. Did you lose any other opportunities due to your participation to the intervention? |
| 3. Has any of your relatives been affected as a result of your participation to the intervention? |

difference in attendance rates between the two phases. This implies that, except the intervention, there has been no special cause of variation in either of the study phases. Owing to the nature of the study, we did not conduct a post-intervention follow-up to observe long-term impact of the intervention because women cannot attend ANC visits after delivery.

## Intervention acceptability

Study findings are summarized in **Table 2**. Of 131 mothers who participated in the program until childbirth (one pregnant woman dropped out), 96.2% were satisfied, 77.1% positively reacted to automated reminders (attitudes), 70.2% expressed willingness to participate, and 86.3% had the ability to actively participate to the intervention. More than half of respondents confirmed that the intervention negatively affected other daily activities as a result of their participation to the program. Overall, 21.4% mothers accepted the intervention. Acceptability of the intervention was higher among mothers who owned a personal mobile phone (p-value 0.0119).

## Discussion

We designed and tested a digital intervention aimed to improve ANC attendance in a rural setting in north-western Burundi. The intervention showed promise of achieving intended effect in the short run. Similar successful evidence of digital-based interventions aimed at improving pregnancy monitoring has been cited in other settings. For instance, Burkinabè women who received digital health-supported ANC at community level appreciated digital interventions and further confirmed that these interventions allow early detection of pregnancy-related complications [27]. Further, community-based digital interventions foster the collaboration

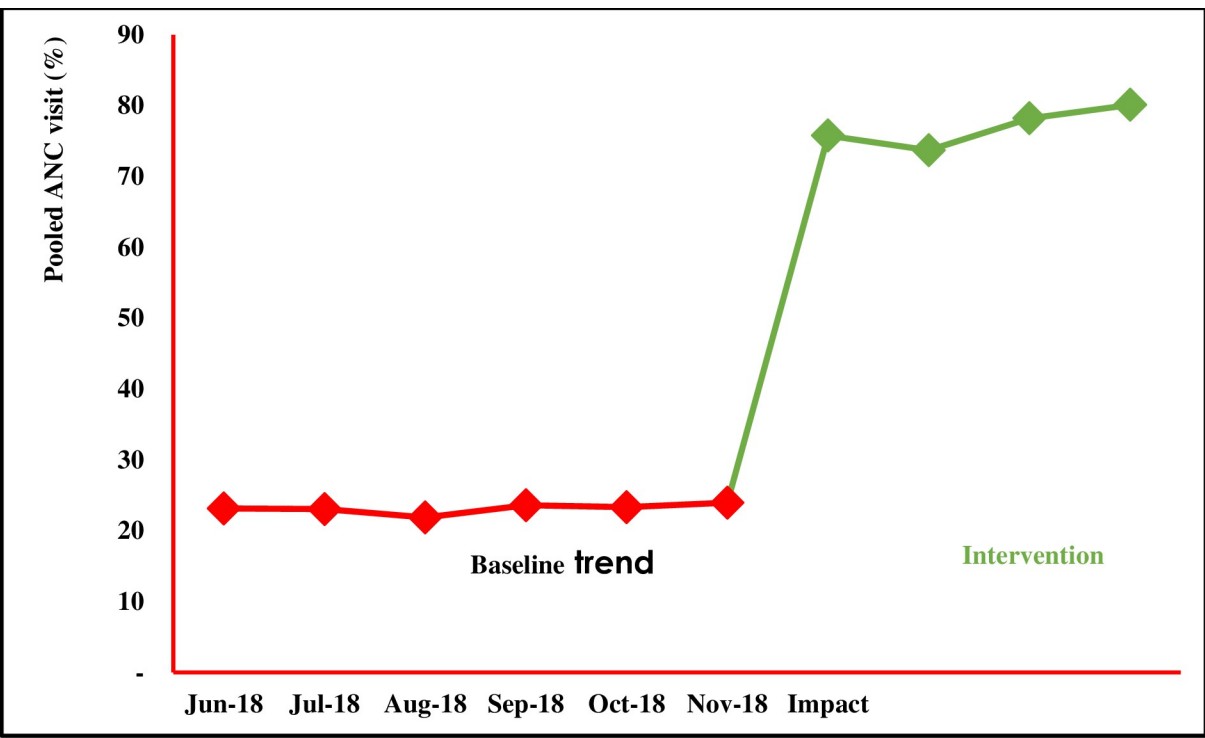

**Fig 4. ANC attendance in Murwi Health Centre before and during intervention.** Fig 4 is a graphical representation of pooled ANC attendance rates before and during intervention. Each datapoint is a mean ANC attendance rate obtained by dividing the total number of women who attended ANC (ANC 1 + ANC 2 + ANC 3 + ANC 4) by 4 and reported as a percentage. Datapoints were plotted on a run chart to observe change in trend before and during intervention. The graph shows an exponential increase of ANC attendance rate as a result of the intervention.

between CHWs and nurses and midwives and are seen as a way to improve collaborative management of the pregnancy while involving pregnant women themselves [27]. In the same perspective, remote ANC monitoring with digital tools improved satisfaction and is believed to constitute a new paradigm of ANC delivery that is safe and cost-effective [28].

In our study, key drivers of success included a multistakeholder engagement and a bottom-up approach. For instance, collaboration and active engagement of CHWs, nurses, and pregnant women contributed to the achievement of intended goals. Successful stories of the project were shared at country, regional, and global level through planned events including the 2019 International Women's Day celebrations in Arusha, and the 7[th] East African Health and Scientific Conference and International Health Exhibition and Trade Fair, Dar es Salaam–Tanzania (March 2019). However, since the idea was solely implemented in a rural setting, its translation to urban contexts needs further exploration. This was strongly recommended by implementing partners and the public. Therefore, the cross-setting evidence of this digital intervention is essential before its translation into routine practice (**Box 1**). Moreover, results from this intervention need to be regarded with caution as this study has some limitations. Firstly, a single-arm study without a control group is not robust enough to confirm attribution of findings. Secondly, we were unable to control the effect of other routine interventions such as community awareness and clinic-based mass awareness campaigns that might have nudged the success of the intervention. Finally, and most significantly, we did not evaluate the study impact on birth outcomes.

Box 1. Quote from a participant to the Dar Es Salaam health conference

*"This is one of the kind technology-grounded initiative in our region that needs to be scaled up. However, since the proof of evidence was generated from a rural setting, it is highly recommended that you pilot the same intervention in a different socio-economic context; I mean an urban and/or semi-urban setting and observe whether the project translates into a similar impact".*

**Rogers Ayiko**, Principal Health Systems and Policy Officer & Head, Reproductive Maternal Newborn Child and Adolescent Health, East African Community, Arusha, Tanzania (during the 7[th] East African Health and Scientific Conference)

**Table 2. Level and determinants of acceptability.**

| | n | percent | $e^{\beta}$ [95% CI] |
|---|---|---|---|
| **Levels of acceptability (n = 131)** | | | |
| Satisfaction | 126 | 96.18 | — |
| Attitudes | 101 | 77.11 | — |
| Willingness | 92 | 70.23 | — |
| Ability | 113 | 86.26 | — |
| Side effects | 71 | 54.21 | — |
| Overall acceptability level | 28 | 21.37 | — |
| **Determinants of acceptability (n = 131)** | | | |
| *Age (ref. group: 15–25 years)* | | | |
| 26–35 years | — | — | 0.43 [0.11–1.65] |
| 36–45 years | — | — | 0.54 [0.08–3.84] |
| *Education (ref. group: none)* | | | |
| Primary | — | — | 1.13 [0.25–5.21] |
| Secondary | — | — | 4.02 [0.75–34.12] |
| *Number of children (ref. group: 1–3 children)* | | | |
| 4–6 children | — | — | 1.04 [0.28–3.82] |
| 6 and more children | — | — | 0.99 [0.74–13.25] |
| *Occupation (ref. group: civil servant)* | | | |
| Farmer | — | — | 4.33 [0.36–51.01] |
| Trader | — | — | 0.94 [0.07–11.90] |
| *Mobile ownership (ref. group: yes)* | | | |
| **No** | **—** | **—** | **0.28 [0.08–0.96]** |
| *Ability to read (ref. group: with difficulties)* | | | |
| Easily | — | — | 0.44 [0.01–25.51] |
| Cannot read at all | — | — | 4.41 [0.53–36.50] |
| *Ability to write (ref. group: with difficulties)* | | | |
| Easily | — | — | 1.62 [0.03–99.61] |
| Cannot write at all | — | — | — |

Table 2 presents results on the level of acceptability by constructs and the overall acceptability. It also includes results from the logistic model on determinants affecting acceptability. Significance is determined by confidence intervals.

## Conclusions

We tested a digital health intervention in a rural setting in north-western Burundi to improve attendance of ANC visits. Early evidence showed a lot of promise in sharply increasing the rate and regularity of ANC visits. We assessed the level of consumer acceptability of the intervention using the acceptability framework adapted from the healthcare interventions acceptability framework by Sekhon et al. (2017) and fitted a logistic model to examine determinants affecting acceptability. Despite the positive impact of mobile-based initiatives on improving ANC attendance, the overall user acceptability is low mostly among women who do not own a mobile phone.

## Acknowledgments

• The Ministry of Health of the Republic of Burundi through the National Programme on Reproductive Health for their collaboration
  • Maternity staff of Murwi Health Center and CHWs for their warm welcome collaboration

## Author Contributions

**Conceptualization:** Nadine Misago, Desire Habonimana, Jean Paul Ndayizeye, Joyce Kevin Abalo Kimaro.

**Data curation:** Nadine Misago, Roger Ciza.

**Funding acquisition:** Nadine Misago.

**Methodology:** Nadine Misago, Desire Habonimana, Jean Paul Ndayizeye.

**Project administration:** Joyce Kevin Abalo Kimaro.

**Resources:** Joyce Kevin Abalo Kimaro.

**Supervision:** Desire Habonimana, Joyce Kevin Abalo Kimaro.

**Visualization:** Roger Ciza.

**Writing – original draft:** Nadine Misago.

**Writing – review & editing:** Desire Habonimana, Roger Ciza, Jean Paul Ndayizeye.

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
