## [Decision Letter · Decision Letter 0]

7 Nov 2022

PDIG-D-22-00271

A Digitalized Program to Improve Antenatal Health Care in a Rural Setting in North-Western Burundi: Early Evidence-based Lessons

PLOS Digital Health

Dear Dr. Misago,

Thank you for submitting your manuscript to PLOS Digital Health. After careful consideration, we feel that it has merit but does not fully meet PLOS Digital Health's publication criteria as it currently stands. Therefore, we invite you to submit a revised version of the manuscript that addresses the points raised during the review process.

Please submit your revised manuscript within 60 days Jan 06 2023 11:59PM. If you will need more time than this to complete your revisions, please reply to this message or contact the journal office at digitalhealth@plos.org. Please include the following items when submitting your revised manuscript:

We look forward to receiving your revised manuscript.

Kind regards,

Danilo Pani, Ph.D.

Academic Editor

PLOS Digital Health

Journal Requirements:

1. Please send a completed 'Competing Interests' statement, including any COIs declared by your co-authors. If you have no competing interests to declare, please state "The authors have declared that no competing interests exist". Otherwise please declare all competing interests beginning with the statement "I have read the journal's policy and the authors of this manuscript have the following competing interests:"

2. Please provide a/amend your detailed Financial Disclosure statement. This is published with the article. It must therefore be completed in full sentences and contain the exact wording you wish to be published.

a. Please clarify all sources of funding (financial or material support) for your study. List the grants (with grant number) or organizations (with url) that supported your study, including funding received from your institution. 

b. State the initials, alongside each funding source, of each author to receive each grant.

c. State what role the funders took in the study. If the funders had no role in your study, please state: “The funders had no role in study design, data collection and analysis, decision to publish, or preparation of the manuscript.”

d. If any authors received a salary from any of your funders, please state which authors and which funders.

3. Please provide separate figure files in .tif or .eps format only and remove any figures embedded in your manuscript file. Please also ensure that all files are under our size limit of 10MB.

4. In the online submission form, you indicated that "Data is available from the first and corresponding author and can be obtained upon valid request". All PLOS journals now require all data underlying the findings described in their manuscript to be freely available to other researchers, either 1. In a public repository, 2. Within the manuscript itself, or 3. Uploaded as supplementary information.

Additional Editor Comments (if provided):

Although this study is interesting from the public health perspective, there are several points of improvement, as suggested by the Reviewers. Writing must be improved for a publication in a Journal paper. Please consider addressing all of them in the manuscrip and in the response letter, providing a rebuttal for the comments you do not want/can accomplish in the revision of the manuscript. Marginal changes to the manuscript cannot be considered a Major Revision, to we will appreciate a substantial revision of your work.

Reviewers' comments:

Reviewer's Responses to Questions

**Comments to the Author**

1. Does this manuscript meet PLOS Digital Health’s publication criteria? Is the manuscript technically sound, and do the data support the conclusions? The manuscript must describe methodologically and ethically rigorous research with conclusions that are appropriately drawn based on the data presented.

Reviewer #1: Partly

Reviewer #2: Yes

2. Has the statistical analysis been performed appropriately and rigorously?

Reviewer #1: Yes

Reviewer #2: Yes

3. Have the authors made all data underlying the findings in their manuscript fully available (please refer to the Data Availability Statement at the start of the manuscript PDF file)?

Reviewer #1: No

Reviewer #2: Yes

4. Is the manuscript presented in an intelligible fashion and written in standard English?

Reviewer #1: Yes

Reviewer #2: Yes

5. Review Comments to the Author

Reviewer #1: I would like to thank the editor for including me in the review process for the manuscript entitled "A Digitalized Program to Improve Antenatal Health Care in a Rural Setting in North-Western Burundi: Early Evidence-based Lessons". The research is important however improving acceptability and allowing for phone ownership may be crucial for increasing effectiveness in the long term. The manuscript is a straightforward attempt that clearly shows the improvement with antenatal care ANC/attendance. The analysis should be further developed on the cross-sectional survey to understand enrolled mothers’ perceptions and acceptability of the digital innovation. The manuscript writing can be improved to reach publication level. Minimal transitions and restructuring can help reader to follow the analysis. The references are appropriate however need to be strengthen with more current research on acceptability, link to intervention design, improvement of antenatal care and birth outcomes. Please find below comments and suggestions for the improvement of the manuscript.

Abstract. Update the sentence with the term “side effects” as it is not fully clear in this context. 

Author’s summary. Add the word “to” improve antenatal care attendance.

Author’s summary. “despite mobile phone ownership” is not clear. 

Page 3, line 7. Since you are mentioning correlation between ANC and childbirth, this strong point needs to be reflected in the subsequent sections below. 

Page 4, line 44. In reference to health outcomes. As stated, the limitations regarding the need for translation into routine practice and evaluation on the impact of health birth outcomes is critical. More supporting evidence with correlations of quality antenatal care, ANC services and attendance to routine practice and birth outcomes should be added to the introduction/background section and structured as part of this manuscript. 

Page 6, line 74. I see that you mention that acceptability determines the uptake of a public health intervention however the results from the theoretical framework of acceptability could go more in depth on the constructs. The level and determinants of acceptability shown in Table 1. could be more robust or expanded on as to understand correlation or impact. Figure 3 (15) can detailed especially since results stated overall user acceptability is very low. It is clear that constructs were measured independently and summed up to yield overall acceptability.

page 7, line 95. The significant finding with 200% increase can be represented in a more meaningful graphic. Figure 4 is clear and straightforward. 

page 9, line 116. Expand or re-word “missing to other revenue generated activities”. This is not fully clear in the context of side effects. 

Page 10, line 138. Regarding translation into routine practice. As stated above, the limitations regarding the need for translation into routine practice and evaluation on the impact of health birth outcomes is critical. I would include more supporting references. 

Page 10, line 140. I agree that a single-arm study without control is not robust enough to confirm attribution of findings however collecting data and nesting a cross sectional survey to understand perceptions and acceptability is very viable. Good for the bottom-up approach being found successful. 

Page 10, line 148. Proof of evidence can be referenced from other studies in the literature. Comparisons of other health interventions and using this data will help further develop the intervention and predict future use.

Reviewer #2: Attention and monitoring for the pregnancy period in Burundi's territories is an issue of particular concern to the public, research, and governments. Careful periodic monitoring of fetal and female health conditions before childbirth can be a useful tool to prevent risk factors and actively act on all known protective factors to promote better health promotion, maternal and newborn health.

Being able to educate, raise awareness and engage even women located in rural areas and consequently more distant from the main urban centers is certainly an excellent screening tool useful to health centers located in the area and, secondarily, promote greater inclusion to urban realities and consequently better development. This is why I have more interest of further exploration in non-urban or at most semi-urban territories than what was suggested by the participant in the health conference, as the rural context is probably still the one at greatest risk.

The ANC guidelines are for this reason suggested by the WHO for a positive pregnancy experience. I found the application of digital technologies (smartphones) to improve access to health interventions very interesting, especially in light of the fact that we are seeing a rapid expansion in the territory of mobile devices that allow better communication with health professionals and emergency channels. 

Thinking about making access to visits more effective by reducing the variable given by pregnant women's forgetfulness of appointments is therefore an excellent health promotion tool. 

However, reading the article led me to ask questions about which clarification and possible further study in the final version would be interesting.

The first item of interest, in my opinion, is how the sample taken in the study was constructed. A reader might be interested to know:

- whether people were contacted on a voluntary basis, 

- whether they were contacted through health facility channels by word-of-mouth or by posting documents, 

- whether they were suggested by nursing staff or other possible modalities. 

It would also be interesting to know whether prior possession of the mobile device was taken into account as a variable. Indeed, it is likely that prior possession (possibly also to be assessed quantitatively in months or years) may positively influence the acceptability of the ANC pathway. A person who has integrated a mobile device (digital friendly) into her daily routine might behave differently than a person who has had it for only a few months or was equipped with the device only on the occasion of this research.

Again with reference to the sample, it would be useful to understand the reasons why some women might not have accepted the intervention. Mention was made of work-care time balance, but were there other possible side effects reported by women? Did the women who did not accept complain about difficulties in accessibility to the device (e.g., difficulty in reading, manual dexterity, using a very small device, etc.) or was it simply very different from their usual communication tools?

The second element of interest is the possibility, which has already been rightly considered in the paper by the researchers, of considering a more complex statistical study that could also control for the prior possession/non-possession variable. The integration and randomization of multiple groups, this time also composed of women who were not in possession of personal device but could be given in use in this research, could give the study an additional element of thinking. Finally, again for the statistical part, it would also be of interest to integrate all the data collected by cPanel. 

The third element I would explore in more detail, however, is qualitative. Line 68 mentions a live chat for direct contact between nurses and women. It may be an interesting data point to understand:

- what kind of digital tool was used for the live chats, 

- whether this device met the elements of usability/accessibility by people, 

- whether people requested this kind of service and in which situations, 

- how many people requested it, 

- what were the topics of the requests made by women, 

- whether the people who provided live chat support had been previously trained in the use of this type of device, and 

- whether the preference for live chat contact was a consequence of the fact that the women on that day were engaged in work activities away from the medical center and thus were able to get information without moving from their place of work or for other reasons (e.g., shyness, shame about live medical personnel, etc...).

An additional qualitative element that might be of interest to the reader is the active or less active participation of the role of the woman's partner, the father of the future child. Does he not participate totally or has some part participated?

A fourth and final element of interest is to understand what kind of adaptation has been made to the acceptability registration questionnaire and whether this instrument has undergone validation from previous research and how Sekhon's model has been readjusted.

I congratulate the colleagues who carried out this interesting research. I hope the comments may have been helpful.

6. PLOS authors have the option to publish the peer review history of their article (what does this mean?). If published, this will include your full peer review and any attached files.

**Do you want your identity to be public for this peer review?** For information about this choice, including consent withdrawal, please see our Privacy Policy.

Reviewer #1: Yes: Lisa Catanzaro, MArch, MPH

Reviewer #2: Yes: Andrea Moi

---

## [Decision Letter · Decision Letter 1]

6 Feb 2023

PDIG-D-22-00271R1

A Digitalized Program to Improve Antenatal Health Care in a Rural Setting in North-Western Burundi: Early Evidence-based Lessons

PLOS Digital Health

Dear Dr. Misago,

Thank you for submitting your manuscript to PLOS Digital Health. After careful consideration, we feel that it has merit but does not fully meet PLOS Digital Health's publication criteria as it currently stands. Therefore, we invite you to submit a revised version of the manuscript that addresses the points raised during the review process.

Please submit your revised manuscript within 30 days Mar 08 2023 11:59PM. If you will need more time than this to complete your revisions, please reply to this message or contact the journal office at digitalhealth@plos.org. Please include the following items when submitting your revised manuscript:

We look forward to receiving your revised manuscript.

Kind regards,

Danilo Pani, Ph.D.

Academic Editor

PLOS Digital Health

Journal Requirements:

Additional Editor Comments (if provided):

According to the Reviewers' comments, the manuscript was improved but there are still some unclear points. I kindly ask the authors to solve, if possible, these issues in order to proceed with the final acceptance.

Reviewers' comments:

Reviewer's Responses to Questions

**Comments to the Author**

1. If the authors have adequately addressed your comments raised in a previous round of review and you feel that this manuscript is now acceptable for publication, you may indicate that here to bypass the “Comments to the Author” section, enter your conflict of interest statement in the “Confidential to Editor” section, and submit your "Accept" recommendation.

Reviewer #1: All comments have been addressed

Reviewer #2: All comments have been addressed

2. Does this manuscript meet PLOS Digital Health’s publication criteria? Is the manuscript technically sound, and do the data support the conclusions? The manuscript must describe methodologically and ethically rigorous research with conclusions that are appropriately drawn based on the data presented.

Reviewer #1: Yes

Reviewer #2: Yes

3. Has the statistical analysis been performed appropriately and rigorously?

Reviewer #1: Yes

Reviewer #2: Yes

4. Have the authors made all data underlying the findings in their manuscript fully available (please refer to the Data Availability Statement at the start of the manuscript PDF file)?

Reviewer #1: Yes

Reviewer #2: Yes

5. Is the manuscript presented in an intelligible fashion and written in standard English?

Reviewer #1: Yes

Reviewer #2: Yes

6. Review Comments to the Author

Reviewer #1: The Authors replied well to all the requests for revisions. Results are clear and updated. In my opinion the paper is ready for publication with minor corrections. 

Page 3, line 7: If the surname of the first author of the referenced article is given, "et al." should be shown with a period. 

Reference section: I am still seeing only 1 article from 2022 and many others are over 5 years old. I am surprised about this in the context of recent technological digital health interventions. 

Thank you for your contributions, updates, and I wish you much success with your research.

Reviewer #2: I would like to thank the authors, the reviewer and the editor for the work done between the original submission and this first review, in which many elements are now clearer.

It is worthwhile, however, to try to elaborate further on some elements so that the paper is even more effective and useful for the scientific community and for people who are doing or will do similar research.

In some places in the paper, a total sample size of 132 women is indicated, while in other places the number 131 appears. Specifically, 132 is indicated in the abstract (seventh line) and in line 85, while 131 is indicated in line 169 and in Table 1 line 176.

Source 20 has a link which is no longer active, it was not possible to elaborate on the statement in line 41, a reader might be interested, in addition to what you have rightly pointed out, in understanding how the relationship between drop out at appointments and forgetting scheduled appointments due to lack of reminders and awareness of the importance of the service was detected.

Line 73. As also suggested by the editor, the presence of the complete data could be helpful for a better reading of the graphs and statistics. In fact, in addition to the graph in figure 1, which shows the distribution according to the four visits, it would be useful to observe the same type of graph in relation to attendance at individual sessions or even the drop-out percentage of people in relation to having attended one, two or three meetings. Knowing, hypothetically, that the biggest drop out was between the first and second meeting, health promotion measures could be put in place to strengthen women more at that stage. It would also be useful to comment further on the reasons why, according to the authors, the ANC2 line in the graph in figure 1 (line 72) shows very low percentages compared to the 3rd (ANC3) and 4th visit (ANC4).

line 116. From the above it would appear that we are referring to the WHO recommendation, which, however, states at least eight meetings (line 31). Instead we believe that the authors' intention is to refer to what was the aim of this research, supported by Habonimana et al. i.e. to ensure 4 or more meetings.

line 129 and 226. We are sure that a possible reader would be interested in specifying how Sekhon's model was repurposed and what reflections led to the development of the constructs of satisfaction, attitudes, willingness, ability and side effects from Sekhon's initial writing cited in References, no. 22. Was the questionnaire readapted by changing the original version? If so, it would be useful for the reader to understand which items were changed and how.

line 132. The process that led to the binary coding of acceptability is not clear. A possible reader might be interested in understanding whether ranges or a cut-off score were used to determine whether acceptability was to be reported as 1 or 0.

line 145. By mistake the word "Antennal" was reported instead of "Antenatal".

line 229. A reader might be interested in a reflection on how, according to the authors, side-effects could influence acceptability.

Thank you for your cooperation.

7. PLOS authors have the option to publish the peer review history of their article (what does this mean?). If published, this will include your full peer review and any attached files.

**Do you want your identity to be public for this peer review?** For information about this choice, including consent withdrawal, please see our Privacy Policy. 

Reviewer #1: Yes: Lisa Catanzaro

Reviewer #2: Yes: Andrea Moi

---

## [Editor Report · Decision Letter 2]

13 Mar 2023

A Digitalized Program to Improve Antenatal Health Care in a Rural Setting in North-Western Burundi: Early Evidence-based Lessons

PDIG-D-22-00271R2

Dear Dr Misago,

We are pleased to inform you that your manuscript 'A Digitalized Program to Improve Antenatal Health Care in a Rural Setting in North-Western Burundi: Early Evidence-based Lessons' has been provisionally accepted for publication in PLOS Digital Health.

Best regards,

Danilo Pani, Ph.D.

Academic Editor

PLOS Digital Health

The authors implemented the requested changes and argumented on the Reviewer's criticisms; the manuscript can be accepted in the present form.